# The Diagnostic Significance of SLC26A2 and Its Potential Role in Ulcerative Colitis

**DOI:** 10.3390/biomedicines13020461

**Published:** 2025-02-13

**Authors:** Lijuan Qian, Shuo Hu, Haizhou Zhao, Ye Han, Chenguang Dai, Xinquan Zan, Qiaoming Zhi, Chunfang Xu

**Affiliations:** 1Department of Gastroenterology, The First Affiliated Hospital of Soochow University, Suzhou 215006, China; zzxmd@126.com (L.Q.); daicg_soochow@163.com (C.D.); 2Department of General Surgery, The First Affiliated Hospital of Soochow University, Suzhou 215006, China; hushuo1996@163.com (S.H.); zhz19950125@163.com (H.Z.); hanyeor@163.com (Y.H.)

**Keywords:** SLC26A2, diagnosis, ulcerative colitis, IL-17 signaling pathway, tight junction

## Abstract

**Background/Objectives:** The solute carrier family 26, member 2 (SLC26A2) gene, which belongs to the family of SLC26 transporters, can be detected in multiple tissues. However, the studies of SLC26A2 in colon-related diseases are still limited and incompletely understood, especially in ulcerative colitis (UC). **Methods:** In this study, we attempted to search and identify putative UC candidate genes within a large number of known genes by multiple bioinformatics analyses. The potential cellular characteristics and biological functions of SLC26A2 in the pathogenesis of UC were also elucidated. **Results:** Notably, SLC26A2 was representative and down-regulated in the intestinal mucosa of patients with active UC, compared to healthy controls. Decreased levels of SLC26A2 were proved to have a more value in diagnosis of UC patients, and closely correlated with some UC characteristics, including the Mayo score and Paediatric Ulcerative Colitis Activity Index (PUCAI). Mechanistically, subsequent results from published datasets and our validated clinical data all strongly implied that SLC26A2 was negatively correlated with the IL-17 signaling pathway, and positively associated with the tight junction, which led to abnormal immune cell infiltration and inflammatory injuries. After establishing the UC mice models in vivo by orally administration of DSS in portable water, SLC26A2 was significantly down-regulated at the mRNA or protein level, when compared to that in the control groups. Furthermore, the correlation analyses confirmed that SLC26A2 was positively associated with CLDN3, and negatively correlated with IL-17A expression in colon tissues. In addition, according to the SLC26A2 expression, UC patients were divided into different subgroups. The potential target drugs for UC treatment, such as progesterone, tetradioxin, and dexamethasone, were initially predicted and exerted anti-inflammatory effects via the common molecule-SLC26A2. **Conclusions:** SLC26A2 might be served as a protective candidate in the UC pathogenesis as well as a potential drug target for UC treatment.

## 1. Introduction

Ulcerative colitis (UC) is a chronic, relapsing, and remitting inflammatory disorder of the colon. It starts from the rectum and can extend proximally to involve the whole colon. Based on the anatomic extent of involvement, patients can be divided into proctitis (E1, disease limited to the rectum), left-sided colitis (E2, distal to splenic flexure) and extensive colitis (E3, disease extends proximal to splenic flexure [1,2]. In recent decades, the incidence and prevalence rates of UC have steadily increased in several regions of the world, especially in developing countries, which have placed a heavy health and economic burden on society [3]. Traditionally, the methods of UC detection include patient-reported symptoms, basic clinical and laboratory tests, as well as endoscopic, histologic and imaging assessments [4,5]. However, those traditionally used methods, such as the most commonly used laboratory markers, including C-reactive protein (CRP), the erythrocyte sedimentation rate (ESR), fecal lactoferrin (FL) and fecal calprotectin (FC), are often nonspecific, inaccurate, or unsatisfactory in the diagnosis and monitoring of this disease [6]. Therefore, searching better biomarkers for predicting the UC prognostication and development are still urgently needed [7].

The solute carrier family 26, member 2 (SLC26A2) gene, which belongs to the family of SLC26 transporters and encodes for a sulfate/chloride antiporter of the cell membrane, is pivotal to the uptake of inorganic sulfate [8]. Previous studies had confirmed that a strong signal for SLC26A2 mRNA and protein immunostaining could be detected in developing fetal hyaline cartilage [9]. Patients with different phenotypes of inherited chondrodysplasias experienced skeletal defects mainly due to SLC26A2 mutations or pathogenic variants, resulting in defective sulfate absorption and uptake in enterocytes and chondrocytes [10]. Furthermore, SLC26A2 expression might also be detected in multiple tissues, such as the eccrine sweat glands, bronchial glands, placental villi, exocrine pancreas, and adrenal cortex [9,11]. As early as 2001, SLC26A2 mRNA expression could be detected in the normal colon, mainly in the absorptive epithelial and goblet cells [12]. Pitule et al. firstly found that SLC26A2 expression was significantly down-regulated in colorectal cancer (CRC) tissues, and patients with a higher expression of SLC26A2 in tumor tissues had a longer overall survival (OS) and disease-free interval (DFI) [13]. However, the studies of SLC26A2 in colon-related diseases are still limited and incompletely understood, especially in inflammatory bowel disease (IBD). In 2009, Comelli et al. identified some novel ileal and colonic genes in Crohn’s disease (CD) and/or UC patients, and SLC26A2 was found to be up-regulated only in Crohn’s colon, but not in Crohn’s ileum or an UC-inflamed ileum and colon, compared to the healthy epithelium [14]. The opposite results of the study by Bjerrum et al. reported that the expression of SLC26A2 was decreased in patients with IBD as having UC or CD [15]. In 2025, a recent study from Zhang et al. firstly focused on nicotinamide adenine dinucleotide (NAD+)-related genes to reveal their potential pivotal roles in UC. SLC26A2 was strongly associated with respiratory electron transport. However, their PCR data for SLC26A2 did not show any significant statistical differences, which could be attributed to an insufficient sample size and differences in patient medication (the authors only selected five UC and five control patients for PCR validation) [16].

Since the first publication two decades ago describing the use of microarray in IBD, this method appears as a good possibility for finding more information on differentially expressed genes and isoforms in UC [17,18,19]. Subsequently, other novel technologies or advances, such as high-throughput next-generation sequencing (NGS) or single-cell RNA sequencing (scRNA-seq) have been applied and transformed biological research, allowing them to collect more accurate and proficient information during the UC initiation and development, and elucidate biological mechanisms even at the cellular level [20,21,22,23]. In this study, we attempted to identify the UC-associated molecules by WGCNA analyses, and evaluate the potential clinical significance of these involved genes in UC colon samples. SLC26A2 was significantly down-regulated in UC colon tissues, and selected for our further experiment. Then, enrichment analyses in GSE87466 and GSE109142 demonstrated the potential relationships between SLC26A2 levels and inflammation signals. Subsequently, using the GSE231993 dataset, multiple bioinformatics analyses at the level of a single-cell RNA-seq further revealed the potential cellular characteristics and biological functions of SLC26A2 in the pathogenesis of UC. According to the SLC26A2 expression, UC patients were divided into different subgroups, and potential target drugs for UC treatment were also initially predicted. In addition, using the UC mice models by oral administration of DSS in portable water, the potential relationships between SLC26A2 and some related molecules of tight junctions (and IL-17 signals) were discussed in vivo.

## 2. Materials and Methods

### 2.1. Data Acquisition and Sample Collection

We obtained the data from the Gene Expression Omnibus (GEO) database (https://www.ncbi.nlm.nih.gov/geo/ (accessed on 22 November 2024) and followed the principles outlined below: (1) Microarray, RNA-seq, and single-cell sequencing data were comprehensively included to ensure reliable results. (2) The dataset for bulk RNA-seq analysis of UC must contain more than 100 samples. (3) The samples of UC must be accompanied by detailed clinical characteristics. For discovery, we selected GSE87466, GSE109142, and GSE56298. For validation, we used GSE92415. The detailed information of the included datasets was is presented in Table 1. The raw data were downloaded, and if not available, the series matrix files were downloaded. The RNA-seq data were normalized to transcripts per kilobase per million mapped reads (TPM) format. In detail, gene exon length information was obtained using the “GenomicFeatures” package (version 1.58.0) in R. Reads per kilobase (RPK) values were calculated using the formula: counts/(exon_lengths/1000). Subsequently, RPK values were scaled by the sample-specific scaling factor and multiplied by one million to derive TPM values.

In addition, 57 colonic mucosa samples from patients diagnosed with UC through digestive endoscopy and pathology were collected. These patients were registered in the Department of Gastroenterology (or General Surgery) at the First Affiliated Hospital of Soochow University from April 2019 to December 2022. Simultaneously, 20 colonic mucosa samples were collected from healthy individuals. Samples were immediately stored in liquid nitrogen upon removal during endoscopy or surgery. The clinical characteristics were collected concurrently. This research work was granted approval by the ethics committee of the First Affiliated Hospital of Soochow University (2022-431) on 20 March 2022.

### 2.2. Weighted Gene Co-Expression Network Analysis (WGCNA)

We performed a WGCNA to identify gene clusters with similar expression patterns, and to identify the most relevant gene clusters associating with UC. Cluster trees of the samples were constructed to exclude any abnormal samples. Subsequently, we selected the top 15,000 genes with a high standard deviation for analysis. The soft threshold power β was determined using the “pickSoftThreshold” function in the R package “WGCNA” (version 1.72.1), which served as the primary tool for this analysis [24]. Genes in the module that showed the highest correlation with UC traits were selected for further analysis.

### 2.3. Identification of Differentially Expressed Genes (DEGs)

We used the R package “limma” (version 3.56.2) to identify genes that were differentially expressed in a statistically significant manner. Genes with an adjusted *p* value < 0.05 and a |logFC| > 2 were selected.

### 2.4. Quantitative Real-Time Polymerase Chain Reaction (qRT-PCR)

RNA extraction from fresh frozen tissues was performed using a TRIzol reagent (Invitrogen, Waltham, MA, USA), and complementary DNA (cDNA) was synthesized using a Prime Script RT Master Mix (Takara, Osaka, Japan). Primers were designed using the “Primer-BLAST” website (https://www.ncbi.nlm.nih.gov/tools/primer-blast/index.cgi (accessed on 20 November 2024)). The reaction and quantification were performed using a FastStart Universal SYBR Green Master (Roche, Mannheim, Germany), and data processing was carried out on a LightCycler 96 System. The expressing levels were quantified using ∆Ct, which meant the average Ct value of target genes minus the Ct value of GAPDH. The primers applied for PCR analysis were listed as follows: ABCB1 (Human): forward, 5′-TTGCTGCTTACATTCAGGTTTCA-3′, reverse, 5′-AGCCTATCTCCTGTCGCATTA-3′; AQP8 (Human): forward, 5′-GCGAGTGTCCTGGTACGAAC-3′, reverse, 5′-CAGGCACCCGATGAAGATGAA-3′; SLC26A2 (Human): forward, 5′-GGCTCCCAAAATACGACCTAAA-3′, reverse, 5′-GGCACCAATAATATGCCCACAA-3′; CFTR (Human): forward, 5′-AAAAGGCCAGCGTTGTCTCC-3′, reverse, 5′-AAACATCGCCGAAGGGCATTA-3′; CLDN3 (Human): forward: 5′- AACACCATTATCCGGGACTTCT-3′, reverse, 5′-GCGGAGTAGACGACCTTGG-3′; CLDN7 (Human): forward: 5′-AGCTGCAAAATGTACGACTCG-3′, reverse, 5′-GGAGACCACCATTAGGGCTC-3′; MYL6 (Human): forward, 5′-GAAGACCAGACCGCAGAGTTC-3′; reverse, 5′-TCCAGCACCTTCACATTCATC-3′; SLC9A3R1 (Human): forward, 5′-GGCTGGCAACGAAAATGAGC-3′; reverse, 5′-TGTCGCTGTGCAGGTTGAAG-3′; GAPDH (Human): forward, 5′-GGAGCGAGATCCCTCCAAAAT-3′, reverse, 5′-GGCTGTTGTCATACTTCTCATGG-3′; SLC26A2 (Mice): forward, 5′-AAGAGCAGCATGACCTCTCAC-3′, reverse, 5′-CTGCCTCAAGTCAGTGCCT-3′; CLDN3 (Mice): forward, 5′-ACCAACTGCGTACAAGACGAG-3′, reverse, 5′-CGGGCACCAACGGGTTATAG-3′; IL-17A (Mice): forward, 5′-TTTAACTCCCTTGGCGCAAAA-3′, reverse, 5′-CTTTCCCTCCGCATTGACAC-3′; and GAPDH (Mice): forward, 5′-AGGTCGGTGTGAACGGATTTG-3′, reverse, 5′-TGTAGACCATGTAGTTGAGGTCA-3′.

### 2.5. Correlation Analysis

We performed correlation analysis between genes and clinical characteristics using the Spearman method, considering *p*-values less than 0.01 as statistically significant. After that, we calculated the scores for the IL-17 signaling pathway and immune cell subtypes using the ssGSEA method, and determined their correlation values using the Spearman method. In addition, a similar correalation method was also employed to reveal the relationships between gene or protein expressions in vivo.

### 2.6. Single-Gene Gene Set Enrichment Analysis (Single-Gene GSEA)

The UC samples were evenly divided into the high and low expression groups based on the expressing levels of the target gene. Gene Set Enrichment Analysis (GSEA) was performed based on the gene set “c2.cp.kegg.v7.2.symbols.gmt” using the R package “clusterProfiler” (version 4.8.2) [25]. Pathways that ranked among the top 50% based on adjusted *p*-values less than 0.05 were considered statistically significant.

### 2.7. Single Sample Gene Set Enrichment Analysis (ssGSEA)

The enrichment scores for the IL-17 signaling pathway were determined using the gene set “IL-17 Family Signaling Pathways” from the “GeneCards” database (https://www.genecards.org/ (accessed on 20 November 2024)). Furthermore, the enrichment scores for 28 immune cell subtypes were evaluated using an annotated gene set downloaded from the “MSigDB” database (https://www.gsea-msigdb.org/gsea/msigdb/index.jsp (accessed on 20 November 2024)). The above enrichment analysis was performed using the R packages “GSEABase” (version 1.62.0) and “GSVA” (version 1.48.3).

### 2.8. Data Processing of Single-Cell Sequencing and Cell Communication Analysis

We used the R package “Seurat” (version 4.3.0.1) to process the data from single-cell sequencing. Firstly, we excluded cells with a gene expression count of less than 200 or more than 7000, and retained cells with a mitochondrial gene expression ratio greater than 20%. Subsequently, we selected the top 2000 highly variable genes using the “FindVariableFeatures” function, and normalized them using the “vst” method. After that, we eliminated the batch effect using the R package “harmony” (version 0.1.1) [26]. Finally, we performed cell clustering using the “FindClusters” function, and assigned labels to the clusters using the R package “singleR” (version 2.2.0) [27]. In addition, we utilized the R package “cellchat” (version 1.1.3) [28], which relied on the “CellChatDB” database, to study intercellular communications between epithelial cells and other cells in the UC colonic mucosa.

### 2.9. High-Dimension Weighted Gene Co-Expression Network Analysis (hdWGCNA)

Due to the sparse matrix nature of the single-cell sequencing data, we employed hdWGCNA to explore the co-expression of genes with SLC26A2 and the specific functions of the gene cluster involving SLC26A2 in epithelial cells. We utilized the R package “hdWGCNA” (version 0.2.19) to accomplish the task [29]. Genes expressed in epithelial cells (at least 5%) were included based on the “SetupForWGCNA” function by setting up the fraction threshold as 0.05. Subsequently, metacells were constructed using the “MetacellsByGroups” function based on the KNN algorithm. The metacells should contain more than 50 cells and fewer than 15 cells shared with others. The optimal soft power value was determined using the “TestSoftPowers” function, which was employed to construct the co-expression network.

### 2.10. Construction of PPI Network

We queried the “STRING” database (https://cn.string-db.org/ (accessed on 20 November 2024)) to explore the interaction network of target proteins, which encompasses a comprehensive collection of known and predicted proteins, as well as their potential relationships [30,31,32,33]. We uploaded proteins associated with SLC26A2, the IL-17 signaling pathway, tight junctions, and other effectors activated in epithelial cells of UC. The results were exported to Cytoscape software (version 3.9.1), and the nodes were ranked based on their “Degree” scores, which were calculated using “Hytohubba” [34]. The darker color indicated a higher “Degree” value, which signified a more prominent position in the network.

### 2.11. Pathway Enrichment and Drug Prediction Analysis

We conducted KEGG pathway enrichment and drug prediction analysis using the online tool “Enrichr” (https://maayanlab.cloud/Enrichr/ (accessed on 20 November 2024)), a comprehensive platform that encompassed over 400,000 terms [35,36,37]. We uploaded the DEGs (|logFC| > 1, adjusted *p* value < 0.05) obtained from comparing samples with high expression levels of SLC26A2 to those with low expression levels. Subsequently, drugs targeting these DEGs were enriched using the “DSigDB” library within the “Enrichr” platform.

### 2.12. Establishment of UC Model In Vivo

UC was induced by oral administration of 3% DSS in portable water. A total of 20 6-week-old male mice (BALB/c, 18–20 g in weight) were divided into the Control (*n* = 10) and UC group (*n* = 10). Mice in the Control or UC groups drank water or 3.0% DSS (35,000–50,000, MW, MP Biomedicals, Santa Ana, CA, USA) for 7 days, respectively. On the seventh day, all mice were sacrificed, and colons were collected from the ileocecal junction to the anus, and opened longitudinally. Half of the colon was fixed in 4% paraformaldehyde for hematoxylin and eosin (H&E) and immunofluorescence (IF) staining; the other half was frozen in liquid nitrogen for qRT-PCR. All experimental procedures were granted approval by the Experimental Animal Care and Use Committee (the First Affiliated Hospital of Soochow University) (2024-658) on 20 December 2024.

### 2.13. Hematoxylin and Eosin (H&E) and Immunofluorescence (IF) Staining

Colon tissues were fixed in 10% neutral buffered formalin. Paraffin-embedded tissues were prepared, cut into sections (4-µm thickness) and stained with H&E. Then, H&E-stained sections were observed by two blinded pathologists, and the histological scores were determined according to the crypt damage, infiltration of neutrophils, and foci of ulceration in the detected colons [38]. For IF staining, colon slides were dewaxed, and the primary SLC26A2 (1:200, PA5-76918, Invitrogen, USA), CLDN3 (1:300, YT0949, Immunoway, San Jose, CA, USA), or IL-17A (1:500, ab302922, Abcam, Waltham, MA, USA) antibody was added for 2 h at 37 °C. Then, the slides were incubated with the secondary antibody (BS10007, 1:100, Bioworld, Tulare County, CA, USA). Finally, the slides were counter-stained with DAPI and imaged. The fluorescence intensity was calculated by Image J software (version 1.54k).

### 2.14. Statistical Analyses

All statistical analyses were conducted using R software (version 4.2.2) and GraphPad Prism 9 (version 9.5.1). The Spearman correlation method was employed for correlation analysis. The data from different groups were compared by a Student’s *t* test. The *p*-values were adjusted using the default Benjamini–Hochberg correction method, with a significance level of 0.05 for all analyses unless otherwise specified. The flowchart was drawn by Figdraw (https://www.figdraw.com/static/index.html#/ (accessed on 20 November 2024)) (Figure 1).

## 3. Results

### 3.1. The Screening of Potential UC-Related Molecules by WGCNA Analyses

WGCNA analyses were conducted on the datasets GSE87466 and GSE109142 after excluding abnormal samples, and the resulting cluster trees were depicted in Appendix A. Soft thresholds of 14 and 18 were selected to achieve a scale-free degree distribution for genes in GSE87466 and GSE109142, resulting in the identification of 30 and 16 modules, respectively (Figure 2A, Appendix A). Subsequently, correlation analysis was performed, revealing that the magenta module in GSE87466 and purple module in GSE109142 exhibited the strongest correlations with UC (Figure 2B). Genes within the magenta module (*n* = 429, cor = 0.78, *p* = 8.1 × 10^−92^) and purple module (*n* = 348, cor = 0.83, *p* = 9.6 × 10^−90^) demonstrated strong correlations with UC, and were chosen for our subsequent studies (Figure 2C). In addition, differential expression analyses were conducted. A total of 134 (85 up-regulated and 49 down-regulated) in GSE87466 and 320 DEGs (242 up-regulated genes and 78 down-regulated) in GSE109142 were screened as possible UC related genes (Figure 2D). Finally, using the Venn analysis, three genes (ABCB1, AQP8, and SLC26A2) were identified through intersection analysis of genes from DEGs and WGCNA modules, and considered as potential UC-associated molecules (Figure 2E).

### 3.2. Clinical Significance of Potential UC-Related Molecules

The expressing levels of these three genes were validated between healthy controls and UC samples. In GSE87466 and GSE109142, the mRNA expressions of ABCB1, AQP8, and SLC26A2 were all significantly down-regulated in UC samples, when compared to healthy controls (Figure 3A,B). Fortunately, the same results were also observed in our collected UC samples, and these three genes were all down-regulated with all *p*-values < 0.0001 (*t*-test) (Figure 3C). Then, the receiver operating characteristic (ROC) curves were used to evaluate the potential usages and values of tissue-derived ABCB1, AQP8, and SLC26A2 as novel biomarkers for UC diagnosis. As shown in Figure 3D, the areas under the curve (AUC) for three datasets (GSE87466, GSE109142, and our validated clinical samples) all exceeded 0.9. These data strongly confirmed the remarkable diagnostic performances of these down-regulated genes in UC diagnosis.

We further examined the relationships between these three UC-related genes (ABCB1, AQP8, and SLC26A2) and some significant clinical characteristics. For instance, Spearman correlation analyses revealed that down-regulated ABCB1, AQP8, and SLC26A2 were all negatively correlated with the Mayo scores as well as the PUCAI scores in UC samples from GSE109142 (Figure 3E,F). However, ABCB1 and AQP8 exhibited poor performance in GSE92415 (*p* > 0.01), and ABCB1 was not associated with the Mayo score according to our self-test statistics (Figure 3G,H). Consequently, we selected SLC26A2 for our further investigations due to its outstanding diagnostic performance and clinical significance.

### 3.3. Enrichment Analysis of SLC26A2 and Its Potential Role in the UC Pathogenesis

Single-gene GSEA analysis was conducted on SLC26A2. Samples were reorganized based on their expressing levels and divided into two groups (SLC26A2 high and low expression) according to the median value. The heatmaps in Figure 4A depicted the top 30 up- and 30 down-regulated genes from GSE87466 and GSE109142, respectively. The DEGs were then utilized for KEGG enrichment analyses. Surprisingly, the IL-17 signaling pathway played a prominent role in both GSE87466 and GSE109142, which was also supported by the subsequent GSEA analysis (Figure 4B,C). Additionally, the GSEA results revealed that SLC26A2 positively or negatively associated with different signaling pathways. Among them, some inflammation- or immune-related signals, such as the IL-17 signaling pathway, NF-kappa B signaling pathway, Toll-like receptor signaling pathway, TNF signaling pathway, and JAK-STAT signaling pathway, were negatively associated with the levels of SLC26A2. However, the tight junction and some metabolism-related signals were positively correlated with SLC26A2 (Figure 4D). These intriguing findings motivated us to further investigate the possible role of SLC26A2 in UC. Next, we took the IL-17 signaling pathway as an example. The scores of the IL-17 signaling pathway in each sample were calculated using the ssGSEA method. Interestingly, the ssGSEA data showed that the scores of the IL-17 signaling pathway were significantly lower in the high SLC26A2 expression group, when compared to the low expression group (Figure 5A). Furthermore, we also observed a strong negative–correlation between the SLC26A2 expressing levels and scores of the IL-17 signaling pathway (Figure 5B). Since SLC26A2 was proved to negatively regulate the inflammation- or immune-related signaling pathways, we speculated that this novel gene might influence the immune cell infiltration during the UC pathogenesis. Thus, a subsequent immune cell infiltration analysis was conducted, and the results revealed that low SLC26A2 levels might lead to more recruitment of immune cells, including different types of innate- and adaptive-immune related cells (Figure 5C). In addition, the scores of involved immune cells were also found to negatively correlate with the SLC26A2 levels (Figure 5D). Based on the above enrichment analyses from GSE87466 and GSE109142, we proposed SLC26A2 as a novel negatively regulated gene in the UC development. Decreased SLC26A2 expression might activate some inflammation- or immune-related signals, and loosen the tight junction between intestinal epithelial cells, finally leading to more immune cell infiltration and inflammatory injuries.

### 3.4. Localization Analysis of SLC26A2 in Single-Cell Sequencing Data During the UC Pathogenesis

Single-cell RNA sequencing (scRNAseq) is a powerful approach for allowing researchers to profile the transcriptomes of thousands of cells simultaneously and study the genetic makeup of individual cells. In GSE231993, the gene count, mRNA count, and mitochondrion ratio in each cell were presented in Appendix A. The number of principal components (PCs) was initially set to 30 based on the elbow plot (Appendix A). Following the computation of “FindNeighbors” and “FindClusters”, cells from eight samples were segregated into 16 clusters based on their principal components, rather than samples, as shown in Figure 6A,B. Subsequently, clusters were signed as T cells, B cell, Epithelial cells, Monocyte, Smooth muscle cells, Endothelial cells, NK cells, and Neurons using the “SingleR” package by comparing cell characteristics with its five internal databases (Figure 6C). As shown in Figure 6D,E, we observed that SLC26A2 was predominantly expressed in healthy intestinal epithelial cells, while its expression was relatively lower in epithelial cells of UC (Figure 6D,E). Thus, we focused on the epithelial cells and conducted further analysis of cell–cell interactions, which revealed numerous differential expression patterns of signaling pathways between healthy individuals and individuals with UC (Figure 6F). In UC epithelial cells, the MIF, APP, and MK signaling pathways were activated, compared to healthy colonic epithelial cells, whereas GALECTIN was inactivated in UC (Figure 6G). Further in-depth analysis showed the relatively activated signaling pathways in UC epithelial cells communicating with others, compared to other cell types, with particular relevance to epithelial cells, monocytes, smooth muscle cells, endothelial cells, and NK cells (Figure 6H). All the above data aroused our interest in understanding the role of SLC26A2 in intestinal epithelial cells during the UC pathogenesis.

### 3.5. High-Dimensional Weighted Gene Co-Expression Network Analysis (hdWGCNA) Revealed the Potential Biological Functions of SLC26A2 at the Level of Single-Cell RNA-Seq

Accordingly, we conducted hdWGCNA to identify the gene clusters, which were associated with SLC26A2, and gained deeper insights into its biological functions in UC. Metacells were generated from a large population of epithelial cells, and genes expressed in more than 5% of cells were selected for WGCNA. The genes displayed a scale-free degree distribution when the soft power was set to 5 (Figure 7A). As a result, these genes were clustered into five distinct modules, along with a grey module in which genes had no co-expressed partners (Figure 7B). As shown in Figure 7C, the brown module, in which SLC26A2 was located, presented significant correlations with “nFeature_RNA” (number of detected genes in cells), “nCount_RNA” (number of detected mRNAs in cells) and “tissue type” (healthy controls or ulcerative colitis). Furthermore, we conducted KEGG pathway enrichment analysis on the genes in the brown module. Notably, the tight junction pathway was once again found to be enriched, as previously observed in the analysis of bulk RNA sequencing data using single gene GSEA (Figure 4D, Figure 7D). The correlation analysis indicated indicates that the brown module exhibited weak correlations with other modules, and the gene scores calculated by UCells in the brown module were highest in epithelial cells, implying a distinct biological function this module possessed (Figure 7E,F). Taking all factors into consideration, our attention was primarily directed towards the IL-17 signaling pathway and the tight junction pathways. Subsequent correlation analysis of bulk RNA sequencing data among genes involved in the tight junction, IL-17 signaling pathway, and immune cells validated our hypothesis. SLC9A3R1, MYL6, CLDN7, and CLDN3, which were involved in the tight junction pathway in the brown module, displayed negative correlations with the IL-17 signaling pathway and the majority of immune cells, while exhibiting a positive correlation with SLC26A2 (Figure 7G,H).

### 3.6. PPI Network Construction and Drug Prediction

In order to elucidate the underlying mechanisms of SLC26A2 in tight junction regulation, we constructed the protein-protein interaction (PPI) network involving SLC26A2, tight junction proteins, IL-17 signaling pathway-related proteins, and activated proteins in epithelial cells from individuals with UC. The network revealed the possibility that SLC26A2 might primarily interact with CFTR, thereby influencing the related proteins, which were involved in the IL-17 signaling pathway and downstream tight junction proteins. Furthermore, the activated proteins in the epithelial cells of individuals with UC also played a pivotal role in the interaction network (Figure 8A). In addition, we determined the expressions of downstream tight junction genes in our collected colonic samples by qRT-PCR. The data demonstrated that the expressions of CFTR, CLDN3, CLDN7, and SLC9A3R1 were all down-regulated in UC tissues, in comparison with healthy controls, except for MYL6 (Figure 8B). Subsequent correlation analysis confirmed that the SLC26A2 levels in 57 UC samples were positively correlated with the expressions of CFTR, CLDN3, SLC9A3R1, and MYL6 (Figure 8C). These data partly re-confirmed SLC26A2 as a tight junction-associated gene during the UC development.

Moreover, we identified drugs that targeted the DEGs between the high and low expressing groups of SLC26A2 in GSE87466 and GSE109142 datasets. The results listed the top 10 potential targeted drugs through SLC26A2. Among them, progesterone CTD 00006624, tetradioxin CTD 00006848, and dexamethasone CTD 00005779 were highly co-enriched (Figure 8D). This meant that these involved drugs might exerted anti-UC effects through the common molecule-SLC26A2.

### 3.7. SLC26A2 Closely Associated with Tight Junctions and IL-17 Signals In Vivo

To further reveal the potential relationships between SLC26A2 and tight junctions (or IL-17 signals) in vivo. Firstly, we established the UC mice models by the oral administration of DSS in portable water, and colon length and histological score were calculated (Figure 9A–D). Then, we determined the mRNA expressions of SLC26A2, CLDN3, and IL-17A in all mice colons. The data showed that SLC26A2 was significantly down-regulated in the UC colon tissues, compared to that in the control groups (Figure 9E). The correlation analysis initially implied that the colon SLC26A2 expression was negatively correlated with the inflammatory injury score, which indicated SLC26A2 as a protective factor in the occurrence of UC (Figure 9F). Subsequent PCR results also showed that the CLDN3 mRNA expression was significantly down-regulated, while IL-17A was up-regulated during the UC development (Figure 9G,H). Interestingly, subsequent correlation analyses also confirmed that SLC26A2 was positively associated with CLDN3, and negatively correlated with IL-17A expression in colon tissues (Figure 9I,J). To further observe the potential relationships between SLC26A2 and CLDN3 (or IL-17A) at protein levels, we performed the IF staining and their corresponding fluorescence intensities of SLC26A2, CLDN3, and IL-17A were calculated (Figure 10A–F). Fortunately, the data was consistent with our results of the PCR analyses, which partly implied that decreased SLC26A2 expression might disrupt the tight junctions between colonic epithelial cells, and positively associated the activation of the IL-17 signaling pathway in vivo (Figure 10G,H).

## 4. Discussion

Despite the progresses in the management of UC over the past few centuries, much less has been achieved in the diagnosis, detection, and monitoring of this disease. Fortunately, we have entered a new era for transcriptome analysis, due to the advances and applications of newly developed technologies, such as microarrays, NGS, next-generation sequencing (TGS), scRNA-seq, and spatial transcriptomics (ST). Many investigators have now noticed the significance of available big data and used multiple analysis methods to disclose the new causes or mechanisms of diverse diseases [39,40,41,42]. Recently, we combined long-read sequencing technology with short-read RNA-seq methods to investigate the transcriptome complexity in CRC. Newly identified splicing isoform TIMP1 Δ4–5 was proved to play a crucial role in mediating the CRC progression, and might be a potential therapy target in CRC [43]. Similarly, we also made attempts at IBD detection. For instance, we successfully used the method of the K-means clustering algorithm to identify human PRKAR2A-derived circRNAs as novel candidates for predicting colitis-associated cancer (CAC). However, the data between UC and healthy controls are not satisfactory, and RTEL1-derived circRNAs had no clinical significance in human UC patients [19]. In this study, to identify putative UC candidate genes within a large number of known genes, differential expression analysis on microarray and RNA-seq datasets were performed from the GSE87466 and GSE109142, including 107 UC and 227 healthy control samples. WGCNA analyses were conducted, and the corresponding modules were selected. The Venn analysis initially considered ABCB1, AQP8, and SLC26A2 as potential UC-associated molecules. This was very interesting and significant. For instance, the ABCB1 expression was strongly diminished in the intestinal mucosa of patients with active UC [44,45]. Similarly, the putative gene AQP8 was also found to be down-regulated in UC tissues, and possibly correlates with the immune cell infiltration of neutrophils, monocytes, and macrophages [46,47]. These published results were consistent with our validated data from the GSE87466, GSE109142, and our collected colon samples.

Notably, in this study, SLC26A2 presented a more statistical significance and clinical value in diagnosis and correlation analysis of some clinical features, including the Mayo score and PUCAI in multiple validated colon samples. However, the potential mechanisms of SLC26A2 in the occurrence and development of UC has not been fully elucidated. Both single-gene GSEA analysis from GSE87466 and GSE109142, and the subsequent KEGG enrichment analyses all implied that the SLC26A2 expression negatively associated with some inflammatory signaling pathways, including the IL-17 signaling pathway, NF-kappa B signaling pathway, Toll-like receptor signaling pathway, TNF signaling pathway, and JAK-STAT signaling pathway. Taking the IL-17 signaling pathway, for example, IL-17 functions as a pivotal pro-inflammatory cytokine predominantly secreted by Th17 cells, and the levels of IL-17 are significantly increased in the intestinal mucosa and serum of active UC patients [48,49,50]. The potential inflammatory effects of IL-17 were partly derived from its synergistic functions with other cytokines and its ability to recruit excessive inflammatory cells, such as neutrophils and the production of chemokines, which in turn intensified the tissue injuries of colon mucosa [51,52]. Fortunately, our ssGSEA data showed that scores of the IL-17 signaling pathway were significantly lower in the high SLC26A2 expression group, and a strong negative correlation between the SLC26A2 expressing levels and scores of the IL-17 signaling pathway was also observed. These results strongly implied that SLC26A2 could be proposed as a protective candidate in the UC pathogenesis, and reduced levels of SLC26A2 might activate the pro-inflammatory cytokines and signaling pathways, finally leading to more immune cell infiltration and inflammatory injuries.

Of course, we need more evidence to support our hypothesis that decreased SLC26A2 expression is a major underlying cause in the initiation and progression of UC. Analyzing the obtained scRNA-seq information from the GSE231993 dataset, SLC26A2 was found to be predominantly expressed in healthy intestinal epithelial cells, compared to other types of cells, such as T cells, B cell, epithelial cells, monocyte, smooth muscle cells, endothelial cells, NK cells, and neurons. Meanwhile, its expressing levels in epithelial cells of UC were significantly decreased, in comparison with healthy intestinal epithelial cells. This meant means that SLC26A2 might play a suppressing role in UC mainly through the intestinal epithelial cells. In addition, the existed existing cell–-cell interactions, and potential communicated signals between epithelial cells and other cell types, were explicitly displayed. To further reveal the putative mechanism of SLC26A2 at the level of single-cell RNA-seq, hdWGCNA was conducted. Our data showed that the brown module, in which SLC26A2 was also involved, was most representative with the features of UC. Importantly, results of KEGG from the brown module implied that the tight junction pathway was closely involved. Previous studies have indicated that tight junctions, adherens junctions, and desmosomes are essential components for the contacts between adjacent intestinal epithelial cells [53]. Tight junctions act as the “kissing points” between neighboring cells, which can form a continuous protective seal to maintain the correct regulation of the intestinal barrier. In the pathogenesis of UC, the intestinal tight junction disruption, including its composition and integrity, can be clearly observed in the inflamed colonic mucosa [54,55,56]. To confirm our hypothesis, we selected the representative genes involved in the tight junction pathway of the brown module, including SLC9A3R1, MYL6, CTTN and CLDN3. The subsequent correlation analyses strongly demonstrated that these tight junction pathway-related genes negatively correlated with the IL-17 signaling pathway and majority of immune cells, while they positively associated with the SLC26A2 expressions. Then, the PPI network and our validated results of PCR analysis all showed that SLC26A2 actually interacted with CFTR, thereby influencing the tight junction-related proteins and IL-17 signaling pathway. Furthermore, we successfully established the mice models by oral administration of DSS in portable water, and found that SLC26A2 was significantly down-regulated in colons, when compared to the control groups. More importantly, correlation analyses showed that SLC26A2 was positively associated with CLDN3, and negatively correlated with IL-17A expression in colon tissues. These data provided us with new evidence of SLC26A2 as a protective factor during the UC pathogenesis in vivo.

We had proved SLC26A2 as a putative biomarker and target in the pathogenesis of UC. According to the SLC26A2 expression, UC patients were divided into different subgroups, and potential target drugs were also initially predicted. Progesterone, tetradioxin, and dexamethasone demonstrated the potential for use in the treatment of UC. The data implied that these three predicted drugs might exert an anti-inflammatory effect through the potential target SLC26A2 gene. In fact, this study has certain limitations. Firstly, the potential and usage of SLC26A2 target drugs for UC treatment is only a speculation. The exact mechanisms remain elusive, which still need more experimental evidence to support in the future. Secondly, the causal relationships between SLC26A2 and tight junctions (or the IL-17 signaling pathway) remain unclear. We still need a series of experiments, such as constructing an SLC26A2 transgenic mice model in vivo and SLC26A2 over-expressing colon epithelial cells in vitro, to reveal its possible upstream or downstream relationships with tight junctions as well as IL-17 signals. Thirdly, our data in this study only included a total of 57 UC patients. An insufficient sample size might affect the statistical analyses of the corresponding clinical data. Finally, our study only included some existing published databases for data mining and analysis. If more data platforms can be added in the future for supplementation, they may provide us with a more in-depth understanding of SLC26A2 in UC.

## 5. Conclusions

In conclusion, our data identified some putative UC candidate genes within a large number of known genes by multiple bioinformatics analyses, including ABCB1, AQP8, and SLC26A2. Notably, SLC26A2 was representative and down-regulated in the intestinal mucosa of patients with active UC, compared to healthy controls. Decreased levels of SLC26A2 were proven to have a more value in the diagnosis of UC patients, and closely correlated with some UC characteristics, including the Mayo score and PUCAI. Mechanistically, the results strongly showed that SLC26A2 was negatively correlated with the IL-17 signaling pathway, and positively associated with the tight junction, which led to abnormal immune cell infiltration and inflammatory injuries during the pathogenesis of UC. Though larger well-designed studies with more clinical and experimental data are required, to the best of our knowledge, this study is the first report to evaluate the potential role of SLC26A2 in UC. SLC26A2 can be served as a novel protective candidate in the UC pathogenesis as well as a putative drug target for UC treatment.

## Figures and Tables

**Figure 1 biomedicines-13-00461-f001:**
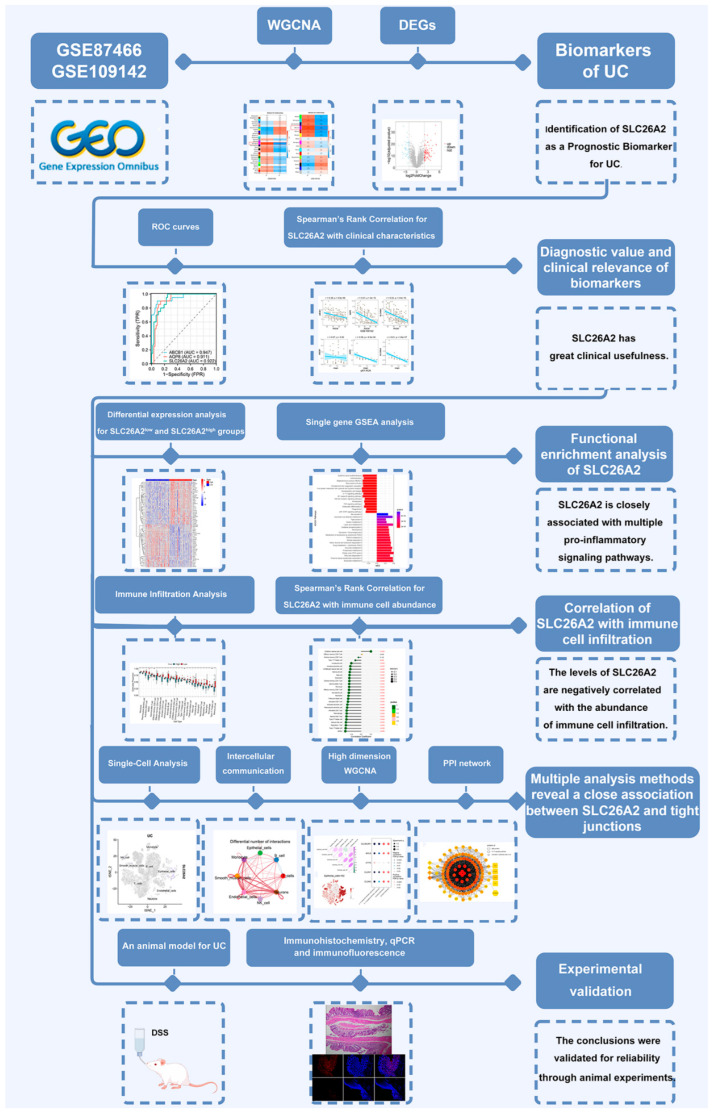
The flowchart of the analysis process.

**Figure 2 biomedicines-13-00461-f002:**
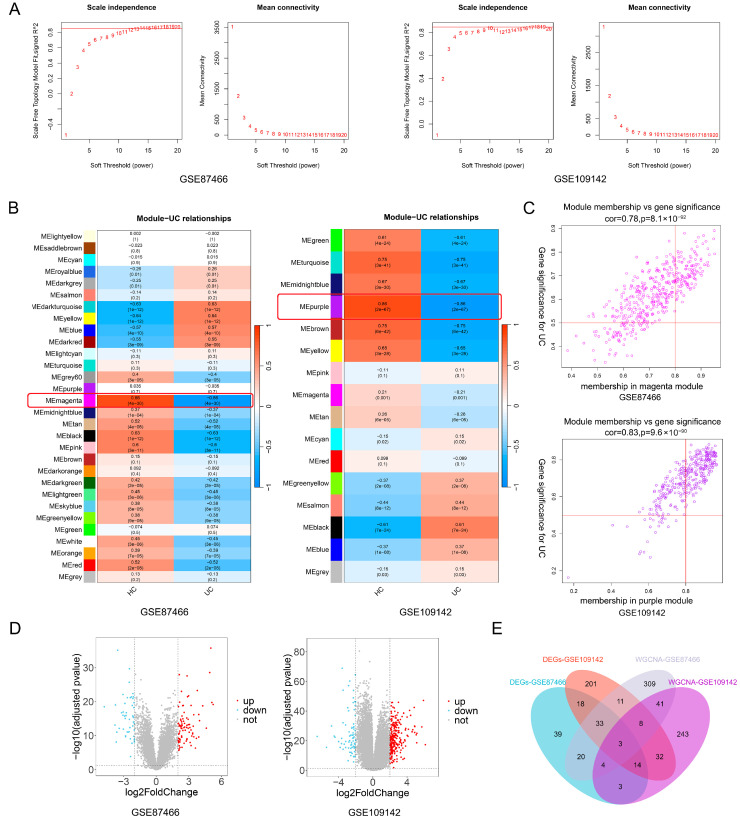
Screening of potential UC-related molecules. (**A**) Selection of the soft threshold for WGCNA (GSE87466 and GSE109142). (**B**) The heatmaps indicated the correlations between modules and UC. Red boxes indicate that magenta module in GSE87466 and purple module in GSE109142 exhibited the strongest correlations with UC. (**C**) The scatterplots showed the relationships between gene significance and module membership in the magenta module of GSE87466 and purple module of GSE109142. (**D**) The volcano plots displayed all the DEGs. (**E**) The Venn diagram illustrates the overlapping genes.

**Figure 3 biomedicines-13-00461-f003:**
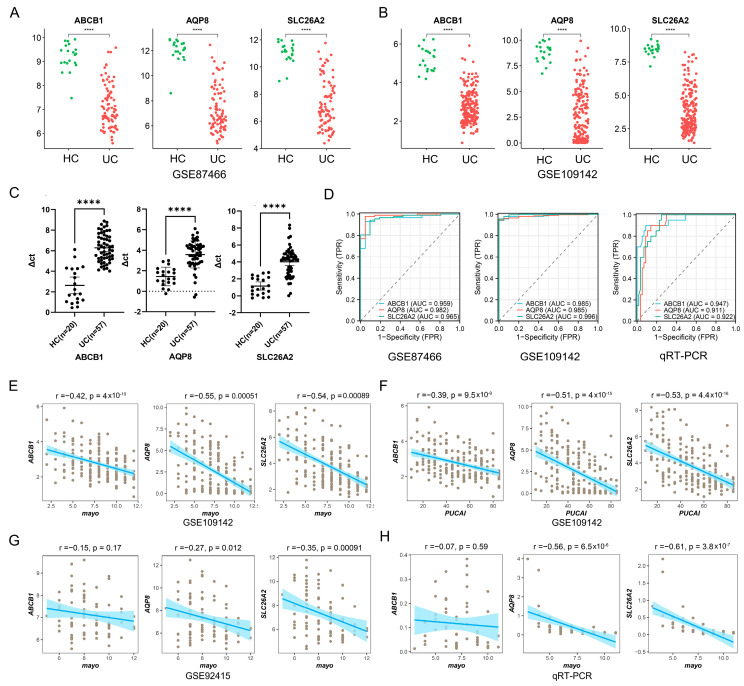
Validate the clinical significance of potential UC related molecules. (**A**,**B**) The boxplots illustrated the expressing levels of 3 UC-related molecules between healthy controls and UC samples in the GSE87466 (**A**) and GSE109142 (**B**) datasets, respectively. (**C**) The ABCB1, AQP8, and SLC26A2 mRNA expressions in our collected UC samples were also determined by the RT-PCR. (**D**) ROC curves were used to evaluate the potential usages and values of tissue-derived ABCB1, AQP8, and SLC26A2 as invasive biomarkers for UC diagnosis. (**E**–**H**) The Spearman correlation analyses revealed the possible relationships between 3 UC-related genes (ABCB1, AQP8, and SLC26A2) and some significant clinical characteristics, including the Mayo and PUCAI scores in UC samples from GSE109142, GSE92415, and our collected UC samples, respectively. (**** *p* < 0.0001).

**Figure 4 biomedicines-13-00461-f004:**
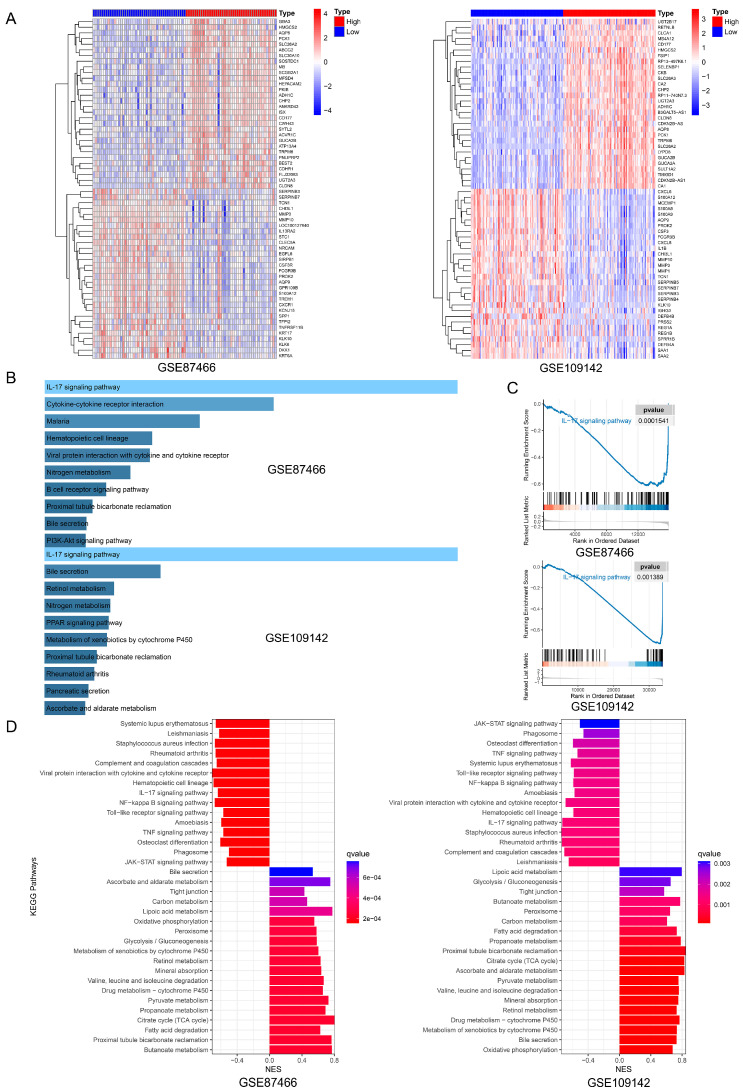
Single gene GSEA analysis was conducted on SLC26A2. (**A**) The heatmaps displayed the top 30 up-regulated and 30 down-regulated genes in the high-SLC26A2 expressed groups, compared to the low-expressed groups. (**B**) The KEGG pathway enrichment analyzed the DEGs in the GSE87466 and GSE109142 datasets. (**C**) The GSEA implied that SLC26A2 was negatively associated with the IL-17 signaling pathway. (**D**) The barplots illustrated the top 50% of enriched pathways in the GSE87466 and GSE109142 datasets.

**Figure 5 biomedicines-13-00461-f005:**
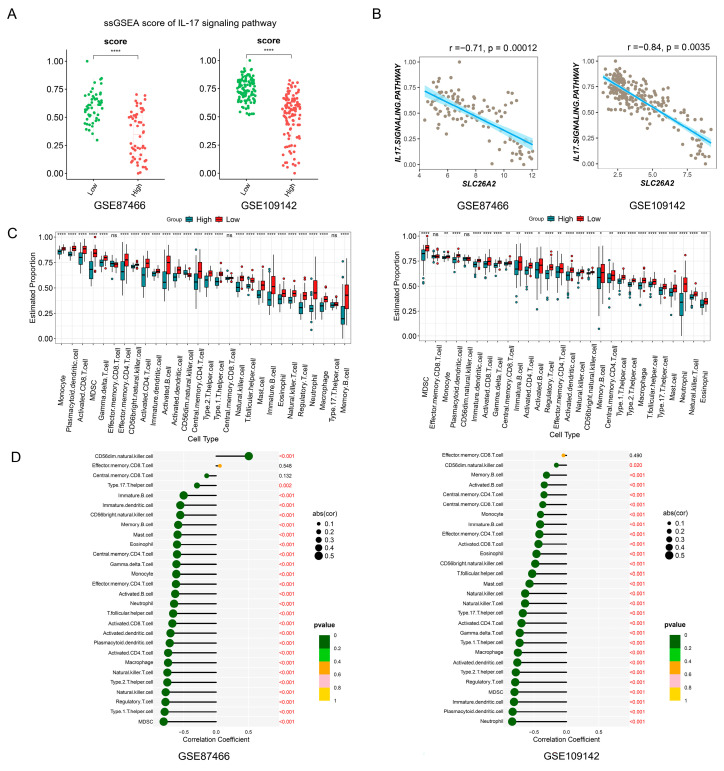
According to the expressing levels of SLC26A2, ssGSEA of the IL-17 signaling pathway and immune cell clusters were performed in the GSE87466 and GSE109142 datasets, respectively. (**A**) The scores of the IL-17 signaling pathway were significantly lower in the high SLC26A2 expression group, compared to the low expression group. (**B**) A strong negative correlation between the SLC26A2 expressing levels and scores of the IL-17 signaling pathway was observed by the Spearman correlation analyses. (**C**) The boxplots illustrated the differences in immune infiltration between the high and low expression groups of SLC26A2. (**D**) The correlations between immune cell infiltration and SLC26A2 were also estimated. (**** *p* < 0.0001).

**Figure 6 biomedicines-13-00461-f006:**
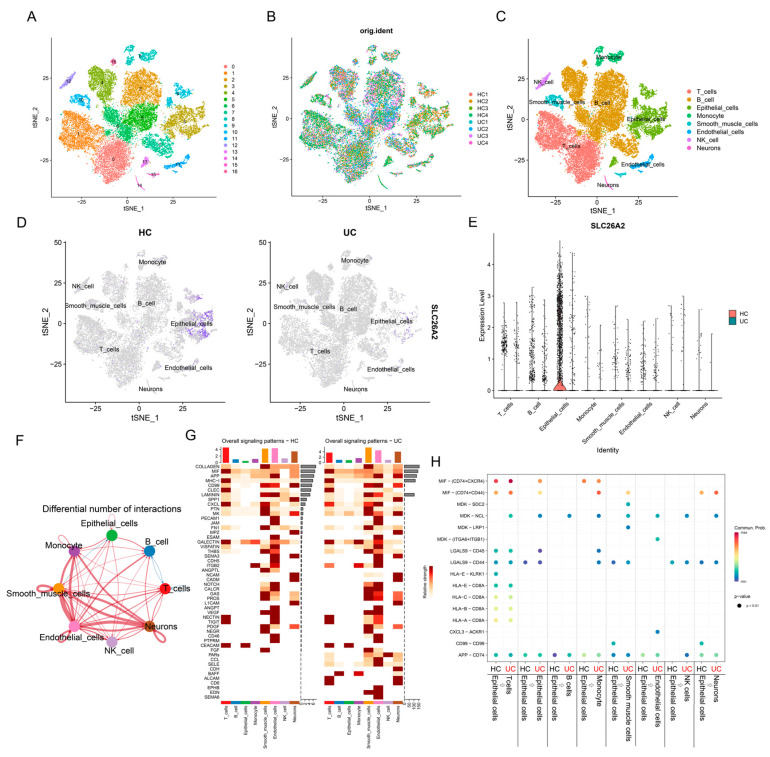
Localization analysis of SLC26A2 in single-cell sequencing data. (**A**–**D**) tSNE plots visualized the cell clusters, original identities, cell annotation, and localization of SLC26A2 in healthy controls and individuals with UC, respectively. (**E**) The distribution of SLC26A2 in cells with annotation was visualized as a violin plot. (**F**) Integrated map of the different numbers of interactions detected between each cell and other cells was presented. (**G**) Intensity demonstration of cell populations in different pathways in healthy controls and individuals with ulcerative colitis. (**H**) Differential signaling pathways in epithelial cells between healthy controls and individuals with UC revealed the diverse biological functions of epithelial cells.

**Figure 7 biomedicines-13-00461-f007:**
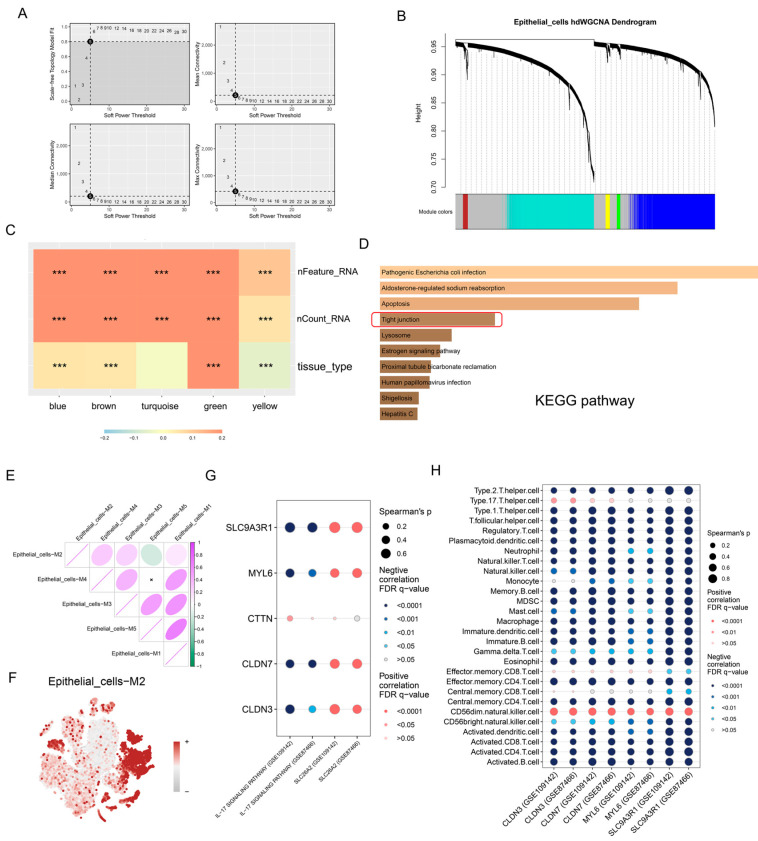
hdWGCNA revealed the potential biological functions of SLC26A2 at the level of single-cell RNA-seq. (**A**) The scale-free topology model displayed the scale-free fit index and the mean connectivity for various soft-thresholding powers. (**B**) The dendrogram illustrated illustrates the different modules in which genes are clustered. (**C**) The correlations between traits and modules were are shown in the correlation heatmap. (**D**) The KEGG pathways enriched by the genes in the brown module were listed, and the tight junction pathway (marked by a red box) was enriched. (**E**) The potential correlations among modules were calculated using the Pearson method. (**F**) Gene scores of the brown module were calculated using the UCell algorithm and presented in each cell. (**G**) The correlations between the genes involved in the tight junction in the brown module and SLC26A2 (as well as the IL-17 signaling pathway), were analyzed using the Spearman method. (**H**) Similar Spearman analyses were conducted between the genes involved in the tight junction in the brown module and immune cell subtypes. (*** *p* < 0.001).

**Figure 8 biomedicines-13-00461-f008:**
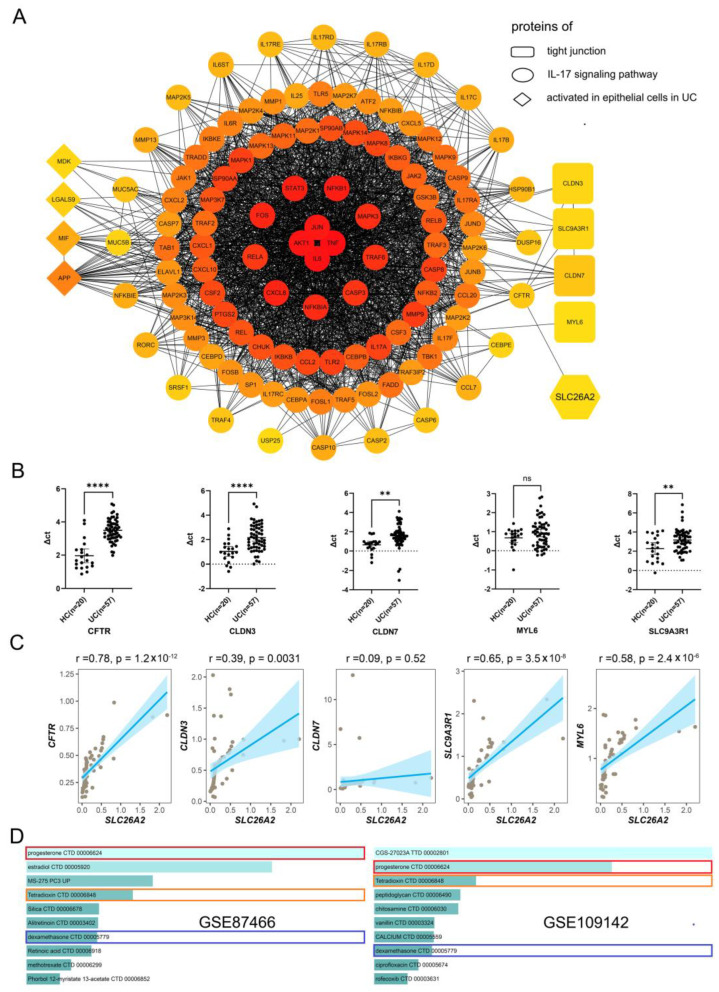
The relationship between SLC26A2 and tight junction genes. (**A**) The PPI network demonstrated the relationships among involved proteins, including SLC26A2, tight junction proteins, the IL-17 signaling pathway, and activated molecules in epithelial cells in UC. (**B**) The mRNA expressing levels of CFTR, CLDN3, CLDN7, MYL6, SLC26A3R1, and SLC26A2 were determined by the qRT-PCR. (**C**) The potential correlations between SLC26A2 and CFTR (CLDN3, CLDN7, MYL6, SLC26A3R1 and SLC26A2, respectively) were analyzed. (**D**) According to the expressing levels of SLC26A2 in GSE87466 and GSE109142 datasets, putative drugs were predicted, and the top 10 drugs were described. Progesterone CTD 00006624, tetradioxin CTD 00006848, and dexamethasone CTD 00005779 were highly co-enriched and indicated by the red, orange and blue boxes respectively. (** *p* < 0.01, **** *p* < 0.0001, and ns: no significance).

**Figure 9 biomedicines-13-00461-f009:**
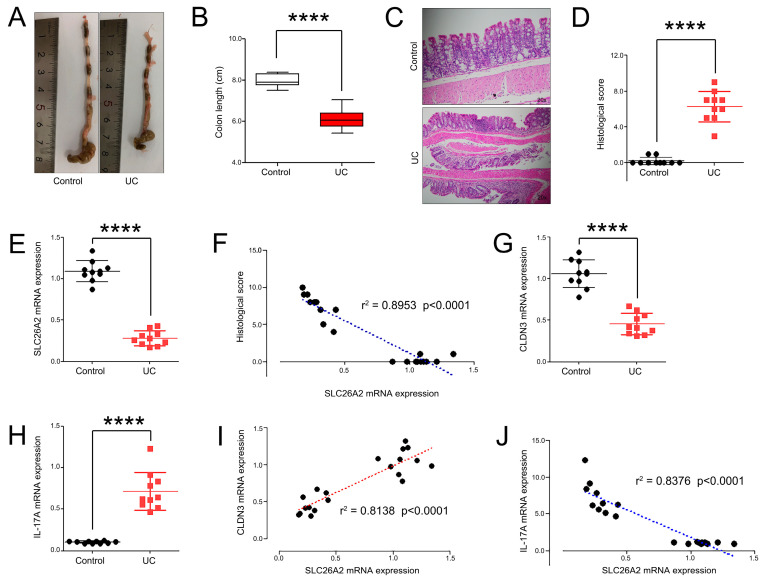
The relationships between SLC26A2 and CLDN3 (or IL-17A) at mRNA levels in vivo. (**A**) Photos of mice colon in each group. (**B**) The colon length of mice on 7th day in the control and UC groups. (**C**,**D**) The colons were stained by H&E, and the corresponding histological scores were compared. (**E**,**F**) The SLC26A2 mRNA expressions in each group were detected by qRT-PCR, and correlation analysis showed the potential relationships between SLC26A2 and histological scores. (**G**–**J**) The CLDN3 and IL-17A mRNA expressions in each group were also determined by qRT-PCR, and the relationships between SLC26A2 and CLDN3 (or IL-17A) were evaluated. (**** *p* < 0.0001).

**Figure 10 biomedicines-13-00461-f010:**
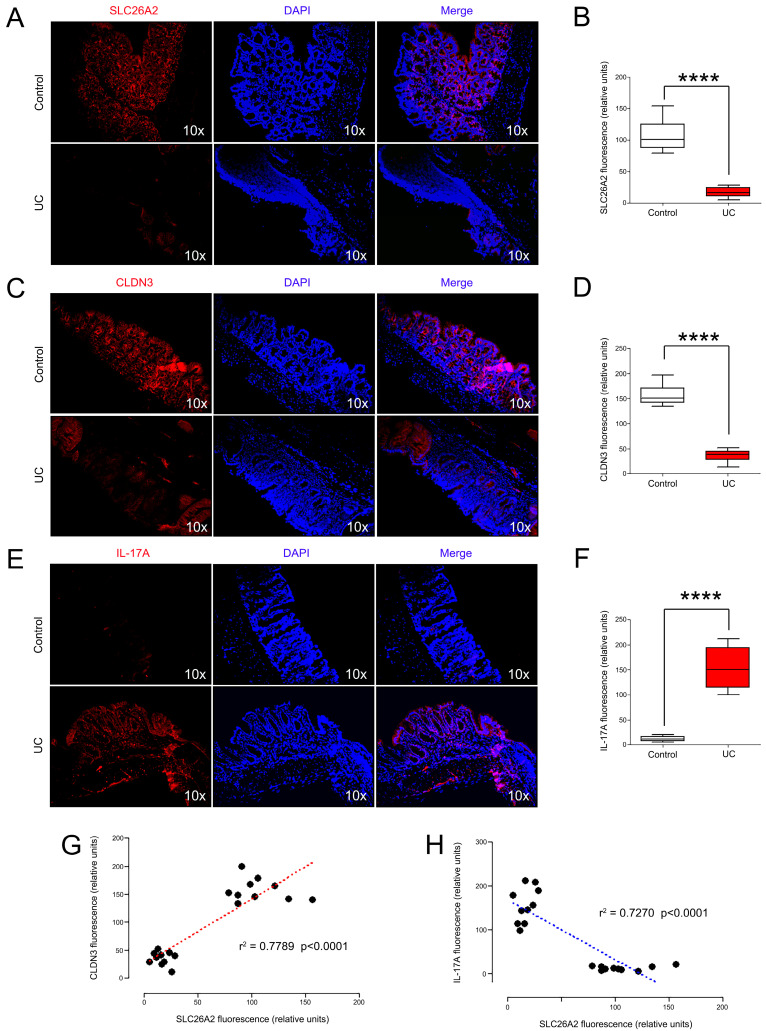
The relationships between SLC26A2 and CLDN3 (or IL-17A) at protein levels in vivo. (**A**–**F**) The protein expressions of SLC26A2, CLDN3 and IL-17A in mice colon were determined by immunofluorescence, and the fluorescence quantification was compared. (**G**,**H**) Spearman–Pearson correlation between SLC26A2 and CLDN3 (or IL-17A) fluorescence intensity in mice was analyzed. (**** *p* < 0.0001).

**Table 1 biomedicines-13-00461-t001:** Data information in this study.

GEO ID	Source	Platform	Participants	Tissues	Attribute
GSE87466	Array	GPL13158	87 active UC and 21 controls	Mucosal	Discovery
GSE109142	RNA-seq	GPL16791	206 active UC and 20 controls	Mucosal	Discovery
GSE231993	Single cell RNA-seq	GPL18573	4 active UC and 4 controls containing 37,967 cells	Mucosal	Discovery
GSE92415	Array	GPL13158	87 active UC with clinical characteristics	Mucosal	Validation

## Data Availability

The datasets (Nos. GSE87466, GSE109142, GSE56298 and GSE92415) generated and/or analyzed during the current study are available in the Gene Expression Omnibus (GEO) database (https://www.ncbi.nlm.nih.gov/geo/ (accessed on 22 November 2024)).

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
