# Peer review of "The Diagnostic Significance of SLC26A2 and Its Potential Role in Ulcerative Colitis"

_biomedicines, 2025, doi:10.3390/biomedicines13020461_

Round 1

Reviewer 1 Report

Comments and Suggestions for Authors

The research article “The diagnostic significance of SLC26A2 and its potential role in ulcerative colitis” by Qian et al is a very interesting and clinically relevant study. The authors have tried to investigate candidate genes in UC by different bioinformatics analyses. The authors have investigated in depth the biological involvement of SLC26A2 in the pathogenesis of UC and have also explored its diagnostic importance. Authors have showed that interestingly, SLC26A2 is significantly downregulated in the intestinal mucosa of active UC patients. The study finding shows that deceased levels of SLC26A2 might be involved in the abnormal immune cell infiltration and inflammatory injuries via involving IL-17 signaling pathway, and the tight junctions. By multiple bioinformatic analysis and further validation in clinical samples, the study indicates that

SLC26A2 might play a direct role in UC pathogenesis and may serve as a potential drug target.  for UC.

The article of high clinical relevance is neat and clear to understand. The study also addresses the limitation of the study.

Bioinformatics analysis performed provide a solid background to the study. The article can be accepted in its current form; however, some minor changes need to be corrected before acceptance.

Minor corrections:

The spelling of “Deceased levels” in the abstract needs to be corrected.

The font and text style vary at some points in the text eg. line 43-44, 234-235.

Author Response

Reviewer 1#:

The research article “The diagnostic significance of SLC26A2 and its potential role in ulcerative colitis” by Qian et al is a very interesting and clinically relevant study. The authors have tried to investigate candidate genes in UC by different bioinformatics analyses. The authors have investigated in depth the biological involvement of SLC26A2 in the pathogenesis of UC and have also explored its diagnostic importance. Authors have showed that interestingly, SLC26A2 is significantly downregulated in the intestinal mucosa of active UC patients. The study finding shows that deceased levels of SLC26A2 might be involved in the abnormal immune cell infiltration and inflammatory injuries via involving IL-17 signaling pathway, and the tight junctions. By multiple bioinformatic analysis and further validation in clinical samples, the study indicates that SLC26A2 might play a direct role in UC pathogenesis and may serve as a potential drug target for UC. The article of high clinical relevance is neat and clear to understand. The study also addresses the limitation of the study. Bioinformatics analysis performed provide a solid background to the study. The article can be accepted in its current form; however, some minor changes need to be corrected before acceptance.

Minor corrections:

The spelling of “Deceased levels” in the abstract needs to be corrected.

The font and text style vary at some points in the text eg. line 43-44, 234-235.

Answer: Thank you very much for your careful and detailed review of our manuscript. Indeed, there were some minor errors in the grammar and layout of some words in our original manuscript. We have carefully read our revised manuscript based on your suggestions, revised each of these grammatical errors one by one, and marked them in red font in the article. For instance, The spelling of “Deceased levels” in the whole text has been corrected. The font and text style in the text eg. line 43-44, 234-235 has been unified. Please check them.

Reviewer 2 Report

Comments and Suggestions for Authors

Dear Authors

Manuscript explains well about some putative UC candidate genes within the large number of known genes by multiple bioinformatics analyses, including ABCB1, AQP8 and SLC26A2.

Minor corrections

1) Please add up-to-date references in the introduction section.

Overall, the solute carrier family 26, member 2 (SLC26A2) was representative and down-regulated in the intestinal mucosa of patients with active UC, compared to healthy controls.

Author Response

Reviewer 2#:

Dear Authors

Manuscript explains well about some putative UC candidate genes within the large number of known genes by multiple bioinformatics analyses, including ABCB1, AQP8 and SLC26A2.

Minor corrections

1) Please add up-to-date references in the introduction section.

Overall, the solute carrier family 26, member 2 (SLC26A2) was representative and down-regulated in the intestinal mucosa of patients with active UC, compared to healthy controls.

Answer: Thank you for your good suggestions. We have reviewed the up-to-date references and added the newly one (39647286, Zhang et. al, 2025) into our revised manuscript. As one of the NAD+-related genes in UC, authors also detected SLC26A2 mRNA expressions in UC. But the expression of SLC26A2 was not significantly different between the UC and healthy controls. Authors only selected 5 UC and 5 control patients for PCR validation. We have added this new references into our introduction section. Thank you for you careful review and good suggestions.

Reviewer 3 Report

Comments and Suggestions for Authors

After reviewed paper: SLC26A2 and ulcerative colitis (UC) demonstrates significant potential but also reveals areas that could be improved. The abstract, while informative, could be better structured by clearly separating the objectives, methods, key findings, and conclusions to enhance readability. Visual representations of the data, though detailed, lack sufficient explanatory text, making them difficult to interpret without prior knowledge of the methods used. 

Methodologically, the study is well-designed but could benefit from a broader dataset selection to reduce geographic bias. Validation through qRT-PCR is a strong point, though additional experimental techniques such as gene knockout or overexpression models in animal studies would provide more robust functional evidence. The use of bioinformatics tools like WGCNA, GSEA, and ssGSEA is commendable, but supplementary pathway analyses through platforms like Reactome or Ingenuity Pathway Analysis could offer deeper insights.

Data interpretation is generally sound, though the study falls short in demonstrating a causal link between SLC26A2 and the IL-17 signaling pathway. The statistical analyses yield very low p-values, raising concerns about potential overfitting or bias that should be addressed. The discussion section speculates on the therapeutic potential of drugs like progesterone and dexamethasone based on bioinformatics predictions, yet lacks supporting experimental validation. 

Finally, the manuscript could be refined through language polishing, particularly correcting minor grammatical issues such as "deceased levels" instead of "decreased levels." Updating older references with more recent studies would also strengthen the paper’s scientific relevance. Including a recommendations section for future research would provide a clearer direction for follow-up investigations. These improvements would enhance the paper's clarity, scientific rigor, and overall impact.

This manuscript should be rejected.

Comments on the Quality of English Language

the manuscript could be refined through language polishing, particularly correcting minor grammatical issues such as "deceased levels" instead of "decreased levels."

Author Response

Reviewer 3#: After reviewed paper: SLC26A2 and ulcerative colitis (UC) demonstrates significant potential but also reveals areas that could be improved. The abstract, while informative, could be better structured by clearly separating the objectives, methods, key findings, and conclusions to enhance readability. Visual representations of the data, though detailed, lack sufficient explanatory text, making them difficult to interpret without prior knowledge of the methods used. 

Methodologically, the study is well-designed but could benefit from a broader dataset selection to reduce geographic bias. Validation through qRT-PCR is a strong point, though additional experimental techniques such as gene knockout or overexpression models in animal studies would provide more robust functional evidence. The use of bioinformatics tools like WGCNA, GSEA, and ssGSEA is commendable, but supplementary pathway analyses through platforms like Reactome or Ingenuity Pathway Analysis could offer deeper insights.

Data interpretation is generally sound, though the study falls short in demonstrating a causal link between SLC26A2 and the IL-17 signaling pathway. The statistical analyses yield very low p-values, raising concerns about potential overfitting or bias that should be addressed. The discussion section speculates on the therapeutic potential of drugs like progesterone and dexamethasone based on bioinformatics predictions, yet lacks supporting experimental validation. 

Finally, the manuscript could be refined through language polishing, particularly correcting minor grammatical issues such as "deceased levels" instead of "decreased levels." Updating older references with more recent studies would also strengthen the paper’s scientific relevance. Including a recommendations section for future research would provide a clearer direction for follow-up investigations. These improvements would enhance the paper's clarity, scientific rigor, and overall impact.

This manuscript should be rejected.

Answer: Thank you for your excellent comments on our submission. We have revised our manuscript and added some complementary experiment to improve the quality of this study. (1) In addition to the clinical data from datasets online, this study also include a total of 57 UC patients from our hospital. These 57 included UC cases contain relevant information, which can help us evaluate the expressing levels and diagnostic role of SLC26A2 in UC. As you know, UC sample collections are more difficult than cancer samples. In our section of “Discussion”, we also have listed some limitations of this study. One limitation is the insufficient sample size, which may affect the results of clinical data. In the future, we will collect more UC colon samples as well as their corresponding clinical information. (2) In our previous submission, we lack of some animal studies to provide more robust functional evidence of SLC26A2 in UC. To initially evaluate the potential mechanisms of SLC26A2, we successfully established the UC mice model, and a series of supplementary experiments partly elucidated that in vivo, decreased mRNA or protein levels of SLC26A2 actually closely correlated with the up-regulated of CLDN3 and abundant of IL-17A (added Figure 9 and Figure 10 in our revised submission). In deed, we still need more experimental approaches, such as building some transgenic or knockout mouse models in animals, as well as conducting the SLC26A2 transfection or interference cells, to clarify the causal relationships these molecules in the future. But to some extent, our added data in Figure 9 and Figure 10 have provided some new evidence in vivo to support our conclusion from the bioinformatics analyses. (3) We have initially predicted some potential target drugs according to the different expressions of SLC26A2. A total of 3 drugs, including progesterone, tetradioxin and dexamethasone, showed possible sensitive according the data of two datasets (GSE87466 and GSE109142). However, we have not deeply investigate the detail mechanisms. The potential and usage of SLC26A2 target drugs for UC treatment is only a speculation. The exact mechanisms remain elusive, which still need more experimental evidence to support in the future. We are plan to further investigate this issue with multiple experiments in vitro and in vivo in the future (The new funding has been received as No.H241466). Besides, we have also emphasized this is a limitation in “Discussion” section for this study. (4) All methods of bioinformatics analyses and statistical analyses have been described and corrected in more detail in our revised manuscript (in red text). (5) Sorry for some minor grammatical issues such as “deceased levels" instead of "decreased levels, we have revised and checked all possible grammar errors in red text. Please check them. (6) Some newly published references and methods-related references have been also added in our revised manuscript.

We would like to thank you for your detailed and helpful suggestions on our manuscript. These excellent comments greatly help us to improve the overall quality of this paper. Thank you.

Reviewer 4 Report

Comments and Suggestions for Authors

The article provides a thorough exploration of the study's findings on SLC26A2 and its potential role in ulcerative colitis (UC). I have some comments and suggestions to improve the focus of this article.

1.       Consider to add 1 figure about the workflow of the study. It can help the readers to understand the whole story of your research.

2.       Section 2.1: Add a justification for the selection of datasets and the TPM normalization method

3.       Section 2.8 and 2.9:

a.       Please provide the reference for all the packages used.

b.       Explain the biological or clinical significance of epithelial cells in UC and how their analysis adds value.

4.       Section 2.10: Provide the reference about string and Cytoscape

5.       Section 2.11: Discuss how pathway and drug prediction results will be used in subsequent experiments or validations

6.       Section 2.12: should be updated. The authors indicate only the programs used but do not describe the statistical methods used in the work

7.       The study hypothesizes that decreased SLC26A2 expression is a major underlying cause of UC and connects this to pathways like IL-17 and tight junction disruption. However, the direct evidence linking SLC26A2 to these mechanisms remains incomplete. What specific experimental data support the role of SLC26A2 in modulating the tight junction pathway and IL-17 signaling beyond correlation analyses?

8.       SLC26A2, as a potential biomarker and therapeutic target, acknowledges that the supporting evidence is largely speculative. What steps are being planned to validate the clinical utility of SLC26A2 as a biomarker? Are the differences in SLC26A2 expression between UC patients and controls sufficient for robust diagnostic or prognostic applications?

9.       The authors mention progesterone, tetradioxin, and dexamethasone as potential drugs targeting SLC26A2, but the mechanisms are described as speculative. What is the evidence for these drugs acting via SLC26A2, and have any in vitro or in vivo validation studies been initiated to confirm this?

10.   The discussion identifies gaps in the understanding of SLC26A2’s mechanisms but does not elaborate on experimental approaches to address them.

Author Response

Reviewer 4#: The article provides a thorough exploration of the study's findings on SLC26A2 and its potential role in ulcerative colitis (UC). I have some comments and suggestions to improve the focus of this article.

  1. Consider to add 1 figure about the workflow of the study. It can help the readers to understand the whole story of your research.

Answer: Thank you for your suggestions. We agree that an added figure about workflow of this study is needed. In our revised manuscript, we have added Figure 1 to help readers better and clearly understand our study.

  1. Section 2.1: Add a justification for the selection of datasets and the TPM normalization method

Answer: Thank you for your detail and helpful comments. The justification for the selection of datasets has been elaborated in the text. The reasons for choosing GSE87466 and GSE109142 are as follows: 1. GSE87466 based on chip technology, while GSE109142 used next-generation sequencing technology. Analyzing these two datasets simultaneously can reduce technical bias (Technical Bias), and using data obtained from two different technologies can enhance the reliability of the results. 2. These two datasets have a sufficiently large sample size (GSE87466 has 87 UC samples, and GSE109142 has 206 UC samples), which can reduce sampling bias, and improve the reliability of the statistical results. 3. GSE87466 and GSE109142 all include detailed clinical data. Meanwhile, we have added the TPM normalization method in the Section 2.1 in red text. Please check it. Thank you.

  1. Section 2.8 and 2.9:
  2. Please provide the reference for all the packages used.
  3. Explain the biological or clinical significance of epithelial cells in UC and how their analysis adds value.

Answer: 1. We have added the related the references for all the packages used in the Section 2.8 and 2.9 of our revised submission. 2. After annotating the cell subpopulations, we explored the expression distribution of SLC26A2 and found that it was mainly located in epithelial cells. On the one hand, one of the main pathological features of ulcerative colitis is the destruction of the epithelial barrier; on the other hand, through intercellular communication analysis, we further found that in ulcerative colitis, there is a stronger communication connection between epithelial cells and other cells, especially within the epithelial cells. This difference is more obvious (Figure 6H). However, this phenomenon is not completely caused by SLC26A2. Therefore, we used hdWGCNA to further explore the possible mechanism of SLC26A2 in epithelial cells. The results also confirmed that the molecules co-expressed with SLC26A2 in epithelial cells may be involved in the process of tight junctions. Since the above conclusions were inferred from the results of cell communication and hdWGCNA, we did not specifically point out this in the Materials and Methods.

  1. Section 2.10: Provide the reference about string and Cytoscape

Answer: We have added the references about string and Cytoscape in Section 2.10. Please check them.  

5.Section 2.11: Discuss how pathway and drug prediction results will be used in subsequent experiments or validations

Answer: In our study, the detail methods of pathway enrichment and drug prediction analysis were described in Section 2.11, and some related references are cited. Our study initially predicted the potential target drugs of SLC26A2 using the “DSigDB” library within the “Enrichr” platform. Nevertheless, this study has some certain limitations, which are discussed in our revised submission. We have emphasized that “The potential and usage of SLC26A2 target drugs for UC treatment is only a speculation. The exact mechanisms remain elusive, which still need more experimental evidence to support in the future”. 

  1. Section 2.12: should be updated. The authors indicate only the programs used but do not describe the statistical methods used in the work

Answer: We have added all statistical methods used in the work. All revisions were done in red text.

  1. The study hypothesizes that decreased SLC26A2 expression is a major underlying cause of UC and connects this to pathways like IL-17 and tight junction disruption. However, the direct evidence linking SLC26A2 to these mechanisms remains incomplete. What specific experimental data support the role of SLC26A2 in modulating the tight junction pathway and IL-17 signaling beyond correlation analyses?

Answer: To elucidate the potential relationships between SLC26A2 and IL-17 (or tight junction disruption). A series of animal experiments in vivo have been done with the recent month. using the UC mice models by orally administration of DSS in portable water, the expressions of SLC26A2, CLDN3 (tight junction-related molecule) and IL-17A (IL-17 signal-related molecule) at mRNA and protein lecels were detected in colons of UC and control mice. And the correlation analyses have been done to reveal the possible association of SLC26A2 with CLDN3 and IL-17A. The results are positive. And the corresponding information of the two added figures (Figure 9 and Figure 10) were described in our revised submission, including the Section of “Introduction”, “Section Materials and methods of 2.12 and 2.13”, “Results of Section 3.7” and “Discussion”. We also emphasized the limitations of added data in the section of “Discussion”. If possible in the future, we will construct the SLC26A2 transgenic mice model in vivo to further reveal its possible upstream or downstream relationships of SLC26A2 with tight junctions and IL-17 signals.

  1. SLC26A2, as a potential biomarker and therapeutic target, acknowledges that the supporting evidence is largely speculative. What steps are being planned to validate the clinical utility of SLC26A2 as a biomarker? Are the differences in SLC26A2 expression between UC patients and controls sufficient for robust diagnostic or prognostic applications?

Answer: Thank you for your comments. In Figure 3D, the diagnostic specificity of SLC26A2 in UC was systematically evaluated by drawing ROC curves based on the clinical data sample information from two databases as well as our collected colon samples. Our data from 57 UC patients was also positive and satisfactory, in which SLC26A2 has a good diagnostic value for UC. Different from the colorectal cancer, the prognosis of UC patients are less mentioned, because of its long-term outcome. Therefore, this study has not focused the prognosis. Because this study only includes 57 UC patients, this is a small-sample study. Now, we are plan to collect more UC colon samples as well as their corresponding clinical information. We hope that the diagnostic role of SLC26A2 with more collected samples may be more satisfactory. 

  1. The authors mention progesterone, tetradioxin, and dexamethasone as potential drugs targeting SLC26A2, but the mechanisms are described as speculative. What is the evidence for these drugs acting via SLC26A2, and have any in vitro or in vivo validation studies been initiated to confirm this?

Answer: Thank you for your good suggestions. In this study, we initially predicted some potential target drugs according to the different expressions of SLC26A2. A total of 3 drugs, including progesterone, tetradioxin and dexamethasone, showed possible sensitive according the data of two datasets (GSE87466 and GSE109142). However, we have not deeply investigate the detail mechanisms. The potential and usage of SLC26A2 target drugs for UC treatment is only a speculation. The exact mechanisms remain elusive, which still need more experimental evidence to support in the future. We are plan to further investigate this issue with multiple experiments in vitro and in vivo in the future. Besides, we also emphasized this was a limitation in “Discussion” section for this study.  

  1. The discussion identifies gaps in the understanding of SLC26A2’s mechanisms but does not elaborate on experimental approaches to address them.

Answer: In our study, we initially showed that SLC26A2 might be closely associated with the tight junction and IL-17A. During the UC pathogenesis, decreased levels of SLC26A2 in colonic epithelial cells might correlate with the breakdown of tight junctions and activiation of the IL-17 signaling pathway. In our previous submission, we still lack of some experimental evidence in vitro or in vivo. To confirm our observations in the bioinformatics analyses and clinical data, we have performed some added experiments in vivo. We successfully established the UC mice model, and a series of supplementary experiments partly elucidated that in vivo, decreased mRNA or protein levels of SLC26A2 actually closely correlated with the up-regulated of CLDN3 and abundant of IL-17A (added Figure 9 and Figure 10 in our revised submission). In deed, we still need more experimental approaches, such as building some transgenic or knockout mouse models in animals, as well as conducting the  SLC26A2 transfection or interference cells, to clarify the causal relationships these molecules in the future. But to some extent, our added data in Figure 9 and Figure 10 have provided some new evidence in vivo to support our conclusion from the bioinformatics analyses.

Round 2

Reviewer 3 Report

Comments and Suggestions for Authors

It may be considered for publication.

Reviewer 4 Report

Comments and Suggestions for Authors

Thank you for your insightful clarification. The article now more clearly and easy to understand 

Comments on the Quality of English Language

Overall is good